# Detecting Unreliable Responses in Generative Vision-Language Models via Visual Uncertainty

## Abstract

Building trust in vision-language models (VLMs) requires reliable uncertainty estimation (UE) to detect unreliable generations. Existing UE approaches often require access to internal model representations to train an uncertainty estimator, which may not always be feasible. Black-box methods primarily rely on language-based augmentations, such as question rephrasings or sub-question modules, to detect unreliable generations. However, the role of visual information in UE remains largely underexplored. To study this aspect of the UE research problem, we investigate a visual contrast approach that perturbs input images by removing visual evidence relevant to the question and measures changes in the output distribution. We hypothesize that for unreliable generations, the output token distributions from an augmented and unaugmented image remain similar despite the removal of key visual information in the augmented image. We evaluate this method on the A-OKVQA dataset using four popular pre-trained VLMs. Our results demonstrate that visual contrast, even when applied only at the first token, can be as effective as—if not always superior to—existing state-of-the-art probability-based black-box methods.

## 1 Introduction

Vision-Language models (VLMs) have seen tremendous growth in recent years exhibiting promising performance across various tasks such as visual commonsense reasoning, image captioning, image retrieval, and object detection. As the capabilities and usage of these multimodal models continue to grow, their tendency to produce incorrect or misleading outputs poses a serious risk, especially in high-stakes settings such as assisting the Blind or Low Vision (BLV) community. Given that VLMs often over-rely on language priors or co-occurring objects in an image Zhou et al. (2023), it is crucial to assess when the model is uncertain about its generation to detect unreliable responses.

The existing uncertainty estimation (UE) methods for VLMs can be broadly categorized into four types: i) *Self-Checking methods:* model evaluates its own correctness via self-evaluation over its generated answer Tian et al. (2023); Srinivasan et al. (2024). ii) *Output Consistency methods:* uncertainty is estimated via examining the consistency of the generated output over multiple question rephrasings Khan & Fu (2024); Shah et al. (2019) or examining model confidence over relevant sub-questions Srinivasan et al. (2024). iii) *Internal state examination methods:* the representation vector of image, question and answer are leveraged to predict the correctness of the response via an MLP-based learnable scoring function Whitehead et al. (2022); Dancette et al. (2023). iv) *Token Probability methods:* these methods use token probabilities to predict the uncertainty Malinin & Gales (2020); Kuhn et al. (2023). In these approaches, the role of visual information on uncertainty estimation remains underexplored especially in black-box approaches, highlighting the need for further research in this direction.

In this work, we investigate the impact of visual information on uncertainty estimation by leveraging visual contrast. Visual contrast, which involves comparing the output token distributions of original and augmented (evidence removed) images, has played a significant role in visual contrastive decoding methods to mitigate object hallucination Leng et al. (2024). However, in this work, we investigate whether it can detect unreliable responses before correcting them especially for open-ended Visual Question Answering (VQA) tasks requiring multimodal understanding, commonsense reasoning and/or external knowledge. Our approach consists of i) two independent forward passes,

one with the unaugmented/original image, and one with the augmented image that has the visual evidence removed, and ii) contrasting the resulting output token distributions at the first generated token. We hypothesize that for unreliable generations, the output token distributions from the original and perturbed images remain similar despite the removal of critical visual information in the perturbed image. We evaluate our method on the A-OKVQA dataset Schwenk et al. (2022), a VQA benchmark that tests the model's external knowledge and commonsense reasoning capabilities. We perform experiments on four popular pre-trained VLMs: LLaVa-7b Liu et al. (2024), LLaVa-13b Liu et al. (2024), Instruct-BLIP Kim et al., and Qwen-VL Bai et al. (2023). Our key contributions are as follows:

- **Investigating visual contrast as an uncertainty metric:** We investigate visual contrast as a measure of uncertainty and find that visual uncertainty can effectively detect unreliable responses in open-ended visual question answering.

- **Analyzing the impact of data augmentations and distance metrics:** We analyze the effect of various data augmentations (e.g., diffusion noise, attention-based masking) and distance metrics (KL divergence, L1, L2) for top 1, 2, and 5 token distributions on uncertainty estimation. Our findings reveal that using a black image and top 1 token-based distance is the most effective approach for detecting unreliable generations.

- **Evaluating SOTA probability-based uncertainty estimators:** We evaluate SOTA black-box uncertainty estimation methods from the LLM literature—Entropy, Cluster, and Semantic Entropy—and find that Semantic Entropy remains a strong uncertainty estimator even for Vision-Language Models (VLMs).

## 2 PROPOSED METHOD

Given a question $q$, and an Image $\mathcal{I}$, a VLM model parameterized by $\theta$ generates an output response sequence $\mathbf{s} = \{s_1, s_2, .., s_k\}$, where $k$ denotes the length of the sequence. In auto-regressive models, the probability of a generated sequence $P(\mathbf{s}|\mathbf{q}, \mathcal{I}, \theta)$ is calculated as the multiplication of its tokens $P(\mathbf{s}|s_{<l}, \mathbf{q}, \mathcal{I}, \theta) = \prod_{l=1}^{L} P(s_l, \mathbf{q}, \mathcal{I}, \theta)$, where $s_{<l} \overset{\triangle}{=} s_l|s_1, s_2, .., s_{l-1}$. Since the product penalizes long sequences, length-normalized confidence acts as an auxiliary probability assigning equal weight to each token $\tilde{P}(\mathbf{s}|\mathbf{q}, \mathcal{I}, \theta) = \prod_{l=1}^{L} P(s_l|s_{<l}, \mathbf{q}, \mathcal{I}, \theta)^{1/L}$ to minimize the impact of length Malinin & Gales (2020). Entropy is another baseline that leverages Monte-Carlo approximation over multiple generated beams $B$, and calculates the entropy approximation as $-\frac{1}{B}\sum_{b=1}^{B} \log \tilde{P}(\mathbf{s}_b|\mathbf{q}, \mathcal{I}, \theta)$ Malinin & Gales (2020). Semantic Entropy is its improved version that clusters semantically similar generations to reduce entropy for consistent/semantically similar generations Kuhn et al. (2023). It sums the scores of all generations belonging to each cluster $\tilde{P}(\mathbf{c}|\mathbf{q}, \mathcal{I}, \theta) = \sum_{\mathbf{s} \in c} \tilde{P}(\mathbf{s}_i|\mathbf{q}, \mathcal{I}, \theta)$, and approximates entropy as $-\frac{1}{|C|} \log \sum_{i=1}^{C} \tilde{P}(\mathbf{c}_i|\mathbf{q}, \mathcal{I}, \theta)$. Cluster entropy is its another variation that counts the number of generations in a cluster and calculates the entropy over normalized counts over clusters Kuhn et al. (2023). Note that entropy and semantic entropy are computationally expensive methods that require multiple beams for good approximation. Self-evaluation is another popular baseline that asks the model itself to evaluate its own generation and uses the confidence of the correctness token as an uncertainty estimate Tian et al. (2023).

In this work, we aim to estimate visual uncertainty by leveraging visual contrast (see overview in Figure 1). For that, we generate an output response $\mathbf{s}$ given the original image $\mathcal{I}$ and question $\mathbf{q}$. For the same question $\mathbf{q}$, we augment the image $\mathcal{I}'$ by removing the visual evidence specific to the question and generate an output response sequence $\mathbf{a} = \{a_1, a_2, .., a_l\}$, where $l$ denotes the length of this augmented image generation sequence. Note that the length of two generations $\mathbf{s}$ and $\mathbf{a}$ can be different and the probability distribution of only first tokens $s_1$, and $a_1$ have the same context such as $P(s_1|\mathcal{I}, \mathbf{q}, \theta)$ and $P(a_1|\mathcal{I}', \mathbf{q}, \theta)$. Due to the context-accumulating nature of these auto-

Figure 1: Illustration of our proposed method, where we compute the distance between the output distributions of first tokens, 'Ted' and 'Lego' generated by the original and the augmented image baselines, respectively.

regressive models, context for the next to-
kens may become different depending on
initial tokens $s_1$ and $a_1$, making the output distributions incomparable. This is because the augmented and original image generate full answer without having token-by-token context alignment as is done in Visual Contrastive Decoding (VCD). In many cases, aligning the initial tokens can lead the augmented image forward pass to guess the answer or next tokens correctly, even when no visual evidence is present. To contrast $P(s_1|\mathcal{I}, \mathbf{q}, \theta)$ and $P(a_1|\mathcal{I}, \mathbf{q}, \theta)$, we experiment with various distance functions on the top 1, 2 and 5 tokens, and find that $|P(s_1|\mathcal{I}, \mathbf{q}, \theta) - P(a_1|\mathcal{I}, \mathbf{q}, \theta)|$ yields the best results on top 1 token. The reference of top tokens is taken from the original image generation.

## 3 EXPERIMENTS AND RESULTS

**Models:** We test UE methods on 4 popular pretrained models. LLaVA-7b, LLaVA-13b, Instruct-BLIP, and Qwen-VL are optimized for visual question-answering task. For the sake of simplicity, we use a single word/short answer prompt for all these models.

**Dataset:** A-OKVQA is a benchmark dataset for open-ended visual question answering that requires multimodal understanding, external knowledge, and commonsense reasoning.

**Evaluation Metric:** To evaluate the accuracy, we use GPT-3.5 model to determine the correctness of the answer given the most probable generation from the model and the ground truth. We record the predicted scores from the UE methods and calculate AUROC (Area Under the Receiver Operating Characteristic), a common metric for binary classifiers. We also record the Prediction Rejection Ratio (PRR), which estimates how well unreliable responses are rejected by a UE method. AUROC scores range from 0.5 (random) to 1.0 (perfect), and PRR ranges from 0.0 (random) to 1.0 (perfect).

**Results:** In our experimental evaluation, we answer the following key questions.

Table 1: AUROC and PRR scores on A-OKVQA dataset

| | | LLaVa - 7b | | LLaVa - 13b | | Instruct-BLIP | | Qwen-VL | |
|---|---|---|---|---|---|---|---|---|---|
| | UE Method | AUROC(%) | PRR(%) | AUROC(%) | PRR(%) | AUROC(%) | PRR(%) | AUROC(%) | PRR(%) |
| AOKVQA | Length-Normalized Confidence | 74.55 | 87.60 | 77.50 | 90.45 | 74.13 | 82.98 | 52.08 | 19.93 |
| | First Token Confidence | 69.39 | 81.30 | 72.96 | 85.30 | 75.09 | 83.47 | 50.00 | 22.89 |
| | Self-Eval Confidence | 71.53 | 79.42 | 63.04 | 87.90 | **76.12** | **84.58** | 43.34 | 19.54 |
| | Entropy | 61.38 | 81.77 | 67.57 | 86.83 | 44.15 | 64.10 | 36.09 | 15.43 |
| | Semantic Entropy | **78.39** | **88.86** | **80.83** | **91.04** | 73.72 | 82.52 | 57.84 | 25.07 |
| | Cluster Entropy | 69.87 | 85.37 | 68.90 | 87.14 | 71.00 | 80.96 | **58.47** | **27.31** |
| | Visual Contrast [First Token] | 75.08 | 87.71 | 74.47 | 89.44 | 71.32 | 82.16 | 51.12 | 23.34 |

***1. Does Visual Contrast contain a reliability signal?*** As shown in Table 1, visual contrast does contain a reliability signal exhibiting comparable AUROC, and PRR scores to existing UE baselines. Specifically, it performs relatively well on LLaVa models, sometimes outperforming strong baselines such as LNC, self-eval, entropy, and cluster entropy. In our visual inspection, we found our metric to be good at rejecting unreliable answers faster at low distance thresholds, which has been the key focus of our hypothesis. However, generations with higher distance values may or may not be reliable, calling for its exploration with other existing baselines, such as ensemble methods.

***2. Which data augmentation method is most effective for unreliability detection?*** As shown in Table 2, we experiment with black image, diffusion noise (T= 900), and attention-based masking, where we mask out the regions of

Table 2: Impact of Image Augmentations on the Visual Uncertainty

| | Black Image | Diffusion Noise | Attention based Mask |
|---|---|---|---|
| UE Method | AUROC(%) | AUROC(%) | AUROC(%) |
| Visual Uncertainty | 75.08 | 71.94 | 57.96 |

high cross-attention between the question and the image in LLaVA-7b model. We find that black-image yields the best performance among these three augmentations.

***3. Which distance metric best captures visual contrast?*** In Table 3, we compare l1, l2 and KL distance to measure visual contrast on top 1, top 2, and top 5 tokens, and find that top 1 token contrast performs the best.

Table 3: Token Distribution Comparison

| | Top 1 | Top 2 | Top 5 |
|---|---|---|---|
| UE Method | AUROC(%) | AUROC(%) | AUROC(%) |
| l1 distance | 75.08 | 72.93 | 70.13 |
| l2 distance | 75.08 | 74.42 | 73.42 |
| KL distance | 75.08 | 69.32 | 65.98 |

**Limitations and Future Directions:** In this work, we explore visual uncertainty aspect of the UE research problem for multimodal models. Since these models integrate both vision and language modalities, visual uncertainty alone might not be sufficient, e.g, for easy questions that model can answer correctly without image, or when output distributions differ despite incorrect answers. Therefore, it would be interesting to explore how combining visual uncertainty with language-based methods could more comprehensively capture both aspects of the problem.

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
