# OpenReview forum: "Detecting Unreliable Responses in Generative Vision-Language Models via Visual Uncertainty"
_ICLR.cc/2025/Workshop/BuildingTrust — Submitted to BuildingTrust_

### Official Review · Reviewer_wgb5 · 2025-02-19
**This paper tackles an interesting problem (uncertainty estimation for VLMs) and suggests that this can be better estimated through applying perturbations to the underlying image. However, I have concerns about how and where these perturbations can be applied.**

**Rating:** 4
**Confidence:** 3

**Review:**

### Strengths:
- This paper centers on improving uncertainty estimation in VLMs. I can imagine many useful applications of this topic and would like to see more research in this area.
- The introduction does a nice job of summarizing the literature. This was a good summary for someone who does not research in this area.
- The results indicate this method achieves similar performance to more computationally intensive methods.

### Weaknesses:
- My main concern with this work is in the selection of visual evidence relevant to the question. If we are using the model itself to select the relevant visual evidence (as is done under the attention-based mask), there is a significant difference between a confidently/unconfidently incorrect model. A confident model may be focusing on the wrong parts of the image and would therefore mask these incorrect parts (in fact, this might help to make it more correct in its outputs). Perhaps this is why the attention-based mask does not perform as well as the black image or diffusion noise. If replacing the image with an all-black image is the plan for the method, this is a much different method that one that *perturbs input images by removing visual evidence relevant to the question*.
- Your results indicate similar performance to the UE method Length-Normalized confidence. What is the advantage of your method?
- In a full length paper, I believe Results sections 2 and 3 would best belong in an appendix. Instead I would have liked to see more discussion about the advantages of your method versus more computationally intensive methods.
- It is not clear to me how GPT-3.5 was used as an evaluator. I could have used more description about this in the evaluation metric section. How accuracy is GPT-3.5 as an evaluator?

### Typos/Suggestions:
- Lines 31, 53: commonsense should be common sense
- Line 86: *improved version* is quite vague, in what sense is it an improvement?
- Formatting on split between page 2 and page 3 is very difficult to read
- Table 2 is likely better represented within text rather than as a table
- I would have liked the results section to have more discussion on Table 1, with less emphasis on Table 2 and 3

---

### Official Review · Reviewer_x1dF · 2025-02-25
**Novel and Cheap UE Method that Could Use Better Use-Case Framing**

**Rating:** 5
**Confidence:** 3

**Review:**

The authors propose a new UE method for unreliable generations that is model agnostic and relatively cheap to compute. I think the general concept of UE and the proposed method are relatively clear, although the related work/literature review is somewhat unclear and hard to get through. The combination of proposed method and lit review makes it hard to follow as well. The proposed method appears to perform decently well/somewhat competitive against other (said to be) expensive baselines, although it does not "win" in any benchmark dataset, which is unfortunate. I think this paper would be stronger if computational runtimes / walk clock times were also compared as that appears to be the edge of this method, for now. Some more analysis on why the top1 token is the most effective would be interesting, and give more intuition on the effectiveness and potential of this method. In summary:

Pros:
- decently performing model that its cheap to commute (and appears to be novel/original)
- straight forward to take away problem context and proposed method's strength from the workshop paper
- UE seems to be pretty significant, especially with rise of LLMs/VLMs, but could use better setting up (see con 1)

Cons:
- proposed model doesn't perform well enough to justify use w/o time comparisons (and better framing that perhaps in some use cases UE has to occur very fast, so computational performance tradeoff is justified)
- more clarity in related works section and intuition for some proposed method choices

---

### Official Review · Reviewer_f3T7 · 2025-03-02
**The paper focuses on uncertainty estimation of vision language models (VLM) and address it using  visual contrast approach. The paper also focuses on effect of distance metrics on AUROC and uncertainty estimators on VLM.**

**Rating:** 6
**Confidence:** 3

**Review:**

Strengths:
1. The paper is easy to read and follow
2. The paper proposes method of uncertainty estimation in VLMs using visual contrast.

Weakness:
1. There is no convincing evidence for the hypothesis that visual contrast is the best uncertainty estimator. In results it is not proving the same hypothesis not for any model, not on average over models. The authors have not experimented on various other datasets for visual question answering.
2. While using the visual contrast approach, the complexity of question is low or easy, the authors can experiment on combination of visual contrast approach and language based uncertainty methods.
3. Intuitive explanation of why black image is better than diffusion noise method for data augmentation.
4.The technical quality of the paper is low

---

### Decision · Program_Chairs · 2025-03-01

Reject